# Enhancing Speech Rehabilitation in a Young Adult with Trisomy 21: Integrating Transcranial Direct Current Stimulation (tDCS) with Rapid Syllable Transition Training for Apraxia of Speech

**DOI:** 10.3390/brainsci14010058

**Published:** 2024-01-06

**Authors:** Ester Miyuki Nakamura-Palacios, Aldren Thomazini Falçoni Júnior, Gabriela Lolli Tanese, Ana Carla Estellita Vogeley, Aravind Kumar Namasivayam

**Affiliations:** 1Department of Physiological Sciences, Federal University of Espírito Santo, Vitória 29040-090, ES, Brazil; 2Superior School of Sciences of the Santa Casa de Misericórdia de Vitória (EMESCAM), Vitória 29045-402, ES, Brazil; aldrentfj@gmail.com; 3Clinic of Speech-Language Pathology, Eldorado Business Tower, Goiânia 74280-010, GO, Brazil; gabi_lolli@hotmail.com; 4Department of Speech and Language Pathology, Federal University of Paraíba, João Pessoa 58051-900, PB, Brazil; anacarlavogeley@gmail.com; 5Department of Speech-Language Pathology, University of Toronto, Toronto, ON M5G 1V7, Canada; a.namasivayam@utoronto.ca; 6Speech Research Centre Inc., Brampton, ON L7A 2T1, Canada

**Keywords:** apraxia of speech, trisomy 21 (down syndrome), transcranial Direct Current Stimulation (tDCS), Rapid Syllable Transition Training (ReST), Broca’s area, Wernicke’s area, supramarginal gyrus, Sylvian temporal parietal junction

## Abstract

Apraxia of speech is a persistent speech motor disorder that affects speech intelligibility. Studies on speech motor disorders with transcranial Direct Current Stimulation (tDCS) have been mostly directed toward examining post-stroke aphasia. Only a few tDCS studies have focused on apraxia of speech or childhood apraxia of speech (CAS), and no study has investigated individuals with CAS and Trisomy 21 (T21, Down syndrome). This N-of-1 randomized trial examined the effects of tDCS combined with a motor learning task in developmental apraxia of speech co-existing with T21 (ReBEC RBR-5435x9). The accuracy of speech sound production of nonsense words (NSWs) during Rapid Syllable Transition Training (ReST) over 10 sessions of anodal tDCS (1.5 mA, 25 cm) over Broca’s area with the cathode over the contralateral region was compared to 10 sessions of sham-tDCS and four control sessions in a 20-year-old male individual with T21 presenting moderate–severe childhood apraxia of speech (CAS). The accuracy for NSW production progressively improved (gain of 40%) under tDCS (sham-tDCS and control sessions showed < 20% gain). A decrease in speech severity from moderate–severe to mild–moderate indicated transfer effects in speech production. Speech accuracy under tDCS was correlated with Wernicke’s area activation (P3 current source density), which in turn was correlated with the activation of the left supramarginal gyrus and the Sylvian parietal–temporal junction. Repetitive bihemispheric tDCS paired with ReST may have facilitated speech sound acquisition in a young adult with T21 and CAS, possibly through activating brain regions required for phonological working memory.

## 1. Introduction

Reduced verbal communicability in people with Trisomy 21 (T21, Down syndrome (DS)) is widely recognized [1], but the nature of these difficulties and effective interventions when co-morbidities are present are relatively unexplored. In a sample of young individuals with T21, Wilson et al. [2] showed that 97.8% met the criteria for motor speech disorders, of which 37.8% showed dysarthria, 22.2% had both dysarthria and childhood apraxia of speech (CAS), and 11.1% had CAS alone. Thus, approximately 33.3% of their sample demonstrated features of CAS. Among those young individuals with T21 who met the criteria for both dysarthria and CAS, 80% demonstrated reduced intelligibility [3].

CAS is defined as a neurological disorder with proximal deficits at the level of speech motor planning and/or motor programming of speech movement sequences [4]. CAS is a difficult-to-treat and persistent motor speech disorder [5,6], and in recent years, there has been a push towards the development, refinement, and standardization of intervention approaches for this condition. In particular, treatment approaches based on principles of motor learning have been suggested with varying degrees of effectiveness and levels of evidence (see [7]).

A promising new approach capable of promoting neuronal plasticity and producing behavioral improvements in motor practice and learning is the use of non-invasive brain stimulation (NIBS) techniques, such as transcranial Direct Current Stimulation (tDCS) and Transcranial Magnetic Stimulation (TMS). These techniques have gained momentum in the last few years and have been shown to promote successful functional recovery after brain injury [8,9]. tDCS has been mostly applied to the post-stroke aphasia population [10,11], due to its feasibility and relatively minor side effects [9,12]. The clinical efficacy of tDCS in this population has been categorized as “possibly effective” (Level C) when applied bilaterally (with the anode over Broca’s area and the cathode over its homologue) [8], but few studies have explored the effects of tDCS on adult apraxia of speech (AOS). Recently, Themistocleous et al. [13] measured the duration of vowels and consonants in spoken words and found that segmental duration was significantly shorter after tDCS over the left inferior frontal gyrus (IFG) along with speech therapy in eight adult AOS patients with non-fluent primary progressive aphasia when compared to the sham condition. These gains were generalized to untrained words and were present 2 months after treatment. Thus, they suggested that tDCS over the left IFG may facilitate speech production in adult AOS patients [13]. However, to our knowledge, no studies have reported the use of tDCS in children with CAS.

Furthermore, typical young adult participants receiving tDCS before performing a nonword task showed significantly greater improvement when compared to participants receiving a sham treatment or those receiving tDCS during a speech learning task [14]. Buchwald et al. [14] suggested that tDCS can improve speech motor learning especially if it is applied immediately before motor practice.

In recent years, there has been a push towards studying the “mode of action” (MoA) through which interventions induce change (e.g., see [15]). Understanding the connection(s) between interventions and the MoA they target would broaden our scientific knowledge on how and why interventions affect change and may result in the development of more effective speech motor interventions (e.g., see [16,17,18]).

In the current study, we used a novel way to identify potential MoAs during speech intervention. Specifically, we measured changes in cortical activation induced by tDCS through scalp electroencephalography (EEG) using event-related potentials (ERPs) [19,20] and focused on the most-studied endogenous ERPs waveform, the P3 (or P300) component [21,22]. According to the hypothesis of “context updating”, the P3 ERP component would reflect the updating of working memory content (see [22]). However, its amplitude will decrease (habituate) when stimuli are repetitively presented and when task performance becomes more automatic [23]. These changes mean that fewer resources are then needed.

In this study, we explore whether repetitive tDCS would facilitate therapy gains in a young adult individual with CAS and T21, and whether this effect would be related to changes on P3 ERP activity in brain regions associated with speech sound production. Given the exploratory nature of this study, we did not make any directional hypothesis regarding brain regions or activation levels for ERP data.

## 2. Methods

This was an N-of-1 randomized trial with a 20-year-old male individual with T21 with moderate–severe apraxia of speech. The research project was approved by the Brazilian Institutional Ethics Review Board of the Federal University of Espírito Santo (CAAE 23866719.8.0000.5060, 13 December 2019) and conducted in strict adherence to the Declaration of Helsinki. It was registered in the Brazilian Registry of Clinical Trials (ReBEC) under the number RBR-5435x9. Informed consent was obtained from the parent and assent was obtained from the participant prior to start of the study.

### 2.1. Participant

To be included in this study, participants must be between 18 to 30 years of age, must present a clinical and/or genetic diagnosis of Trisomy 21 and fulfill criteria for intellectual disability (*Diagnostic and Statistical Manual of Mental Disorders*—fifth edition, DSM-5) or intellectual development disorder (*International Classification of Diseases* 11th Revision, ICD 11) [24]. They must also meet criteria for communication disorders (DSM-5), more specifically for speech sound disorder. Finally, participants were only included if they were able to understand spoken Brazilian Portuguese and/or follow simple instructions and provide consent to participate.

Participants were excluded if they were diagnosed with other mental disorders (e.g., attention deficit hyperactivity disorder, depressive and bipolar disorders, anxiety disorders, autistic spectrum disorder, and impulsive disorders), presented with severe and profound intellectual disabilities, had a history of epilepsy, severe head trauma, presence of implanted devices (e.g., cochlear implants/cardiac pacemaker or intracardiac metal lines), and if they had any metal in their brain or skull (splinters, fragments, pins, etc.). Furthermore, participants must be clinically fit for the treatment proposed in the study, and, therefore, must not have past or current illnesses that could be aggravated during treatment (e.g., neurodegenerative diseases, respiratory illness, etc.).

### 2.2. N-of-1 Study Design

The N-of-1 study design is often used to investigate the effects of treatments for subjects presenting with unique conditions. It has been used to investigate the effects of neurorehabilitation [25,26] in communication disorders [27,28]. In N-of-1 designs, the randomization of treatment allocation at multiple points in time is introduced, allowing for the application of parametric statistical analyzes like those used in randomized clinical trials employing groups of multiple subjects [25,27,28,29]. According to Margon and Giuliano [30] and Samuel et al. [31], N-of-1 trials use many concepts from randomized controlled trials (RCTs), reducing the possibility of bias and, therefore, increasing the validity of findings. Samuel et al. [31] mentioned that evidence based on the N-of-1 trial methodology, also called personalized trials, has increased rapidly in the last decade.

### 2.3. Experimental Procedures

As this study took place during the 2020–2021 global pandemic, strict COVID-19 health protocols were followed (adequate masks, sanitization, and physical distancing—only the experimenter and the participant were allowed in the experimental set) for the duration of this study.

The participant underwent a non-invasive brain stimulation paradigm using tDCS or sham-tDCS in conjunction with a motor speech intervention known as Rapid Syllable Transition Training (ReST) (Figure 1).

#### 2.3.1. Non-Invasive Brain Stimulation

A portable tDCS device (1 × 1 mini-CT, Model 1601-LTE, Soterix Medical Inc., New York, NY, USA) was used to deliver tCDS. Ten tDCS sessions (current intensity of 1.5 mA, electrode size of 25 cm^2^, anode over F5 (Broca’s area—BA44/45), and cathode over F6 (right contralateral region)) and ten sham-tDCS sessions were randomly (www.randomizer.org) distributed in blocks of 2 to be applied as one session per day every other day (three times a week). To maximize cortical effects, they were administered in two 13 min applications with a 20 min interval (13:20:13 protocol) [32,33]. During both 13 min tDCS/sham-tDCS applications, the participant was kept seated at rest (i.e., free to listen to music or watch short movies of his choice on his mobile phone). During the 20 min interval, the training (i.e., pre-practice) phase of ReST was applied remotely (see the description below), and the ReST practice phase was conducted immediately after the second 13 min of tDCS or sham-tDCS application (Figure 1).

#### 2.3.2. Speech Intervention

ReST [32] is a speech intervention based on principles of motor learning and has been recommended for CAS. ReST aims to improve speech production and prosody through training nonsense words (NSWs) with varied stress patterns. Due to limitations to face-to-face delivery during the COVID-19 pandemic, this intervention was chosen in part for its suitability for remote tele-health administration. ReST has sufficient data to warrant its use for children with CAS in both in-person and tele-health formats [34,35,36].

In the current study, trisyllabic NSWs with two stress patterns (strong–weak–weak or weak–strong–weak) were used (e.g., gótabe, faduque). All NSWs used in the ReST treatment were balanced according to the patient’s inventory of sounds and level of motor complexity (based on mandible–lip–tongue movement transitions (e.g., [37])) and met the phonotactic constraints of Brazilian Portuguese words. These NSWs were checked and validated by two licensed Brazilian linguists. A licensed speech–language pathologist (ACEV), blind to the treatment conditions, remotely provided the ReST intervention and presented the NSWs to the participant during the training (pre-practice) phase of ReST. The speech–language pathologist was formally trained to administer ReST with fidelity in Brazilian Portuguese.

For the practice phase of the ReST, the target NSW utterances employed in the pre-practice phase were pre-recorded and randomly presented via a computer using Presentation^®^ software (Version 18.0, Neurobehavioral Systems, Inc., Berkeley, CA, USA, www.neurobs.com) in a quasi-automatable way. Written NSWs were shown as pictures with diagrams cueing the strong syllable (Figure 2) simultaneously with pre-recorded audio (all trisyllabic NSWs were pre-recorded by the speech–language pathologist) of approximately 1000 milliseconds in duration. Each NSW presentation lasted for 20,000 milliseconds. An interval of 2000 milliseconds was interposed between each NSW presentation with a default screen consisting of a black background with a small yellow cross mark in the center to keep the subjects’ attention on the screen. The practice phase lasted approximately 20 min, during which the participant repeated each of the 10 NSWs 5 times for a total of 50 instances of speech production per session. The speech production of each practice phase of ReST was recorded using OBS studio 30.0.2 software and transcribed offline.

Ten NSWs were randomly chosen for tDCS, and another set of ten different NSWs was chosen for the sham-tDCS condition. These 10-NSW sets were kept constant over 10 sessions of each condition to facilitate speech motor practice and learning. Within each session, the 10 NSWs were randomly presented. A third set of ten other NSWs constituted the control condition, which was tested on 4 days randomly distributed across NIBS sessions when no sham-tDCS or tDCS applications were conducted. These control sessions were free of brain stimulation procedures, allowing the researchers to verify the potential occurrence of any placebo effect when comparing them to the sham-tDCS condition. Finally, a fourth set of ten different NSWs was used as a probe and applied at the beginning (initial) and at the end (final) of the study protocol. This allowed for the verification of any effects related to the repeated presentation of a NSW set. Each session was about 90 min in duration and was carried out every other day (~3 sessions per week, over the 10 weeks), for a total of 26 sessions. Although it is possible to run two tDCS training sessions in a day, we only ran 1 session per day due to logistics.

Among the different parameters analyzed in the ReST treatment program (sounds, beats, and smoothness), speech sound accuracy during NSW production and speech sound production was chosen as the main outcome for this study, as it seemed to be the most representative of the participant’s efforts in motor programming and planning to pronounce the NSWs. Additionally, this parameter could be objectively extracted, as all three syllables in a trisyllabic NSW must be produced correctly for the utterance to be scored as correct and computed as 1, and 0 (zero) was computed when one, two, or all syllables were incorrectly pronounced, as recommended by the ReST therapy data sheet (https://rest.sydney.edu.au/ (accessed on 4 February 2021)). The mean percentage (%) of correct responses (±standard error of the mean—SEM) was calculated for each ReST practice session considering 50 trisyllabic NSW utterances in each practice session (10 NSWs repeated 5 times each).

#### 2.3.3. Speech Assessments

The following tests were remotely administered: (1) The ABFW Child Language Test (ABFW) is used to test phonology, vocabulary, fluency, and pragmatics. It was created and validated for the Brazilian child population [38]. The vocabulary evaluation consists of nine different semantic fields (clothing, animals, food, transportation, furniture and fixtures, professions, sites, shapes and colors, toys, and musical instruments), providing percentage scores. The phonological test consists of 34 pictures of objects for naming and 39 words for imitation. From this test, the correct consonants can be counted, and the PCC index can be calculated [39]. (2) The Montreal–Toulouse Language battery (Brazilian version; MTL-BR) was used for the evaluation of language comprehension [40,41]. This test assesses spoken and written language, praxis, and arithmetical skill [41]. (3) The FOCUS-34 parent and clinician tool [42] (Brazilian Portuguese version) is designed to measure functional outcomes in everyday life. Additionally, the intra-session consistency of production was assessed through examining the number of correct repetitions of NSWs. All speech outcome measures were double-checked for reliability (no errors or disagreements were present).

#### 2.3.4. Event-Related Potentials

Electrophysiological event-related potentials (EEG/ERPs) were recorded during the ReST practice phase of each session (Figure 1) through a 30-channel wireless system powered by a lithium battery and with dry electrodes (Quick-30, Cognionics Inc., San Diego, CA, USA) (Figure 2). Electrodes were placed over the scalp according to the international 10/20 EEG system. Data were recorded with a sampling rate of 500 Hz, filtering between 0.5 Hz and 100 Hz with an auricular electrode (A1) as a reference, and having NSWs presented during ReST practice as stimuli.

EEG data were post-processed using BrainVision Analyzer 2.1.2 Professional software (BrainProducts GmbH, Munich, Germany) (Figure 2). Data were filtered from 0.5305164 (order 2, time constant 0.3) to 30 Hz (order 2) with the notch enabled at 60 Hz. Ocular correction was carried out through an independent component analysis, with the Fp1 channel as a blink marker. Next, artifact removal was inspected semi-automatically. Finally, all datasets were segmented into epochs from −200 to 1000 ms relative to picture and audio onset and averaged. All epochs were retained. Control correction was performed using the pre-stimulus interval (i.e., −200 to 0 ms). Low-resolution brain electromagnetic tomography analysis (LORETA) was applied to estimate the three-dimensional intracerebral current density distribution (μA/mm^2^) [43,44,45,46].

Current source densities (CSDs) of the P3 segment were measured within the interval between 250 and 350 milliseconds (Figure 2) in regions of interest (ROIs). We specifically extracted measurements from regions related to speech and from regions surrounding the tDCS electrodes’ position: Broca’s area (left BA 44/45), right contralateral region (right BA 44/55), Wernicke’s area (left BA 22), Sylvian temporal–parietal junction (left BA 22/39), left and right supramarginal gyrus, left and right inferior parietal lobule, left and right dorsolateral prefrontal cortex (left and right BA 9/46), left and right frontal eye field (left and right BA 8), left and right ventrolateral prefrontal cortex (coordinates—left: −32, 56, 6; right: 34, 54, −4; radius: 10 mm [47]), ventromedial prefrontal cortex (coordinates: −2, 32, −10; radius: 10 mm [48,49]). The brain activity in these different regions was then compared between treatment conditions (tDCS vs. sham-tDCS) and within therapy sessions and correlated to sound production during ReST practice performance.

#### 2.3.5. Statistical Analysis

SPSS Statistics Base 24.0 (SPSS Inc., Armonk, NY, USA) and GraphPad Prism 7.0 (GraphPad Software Inc., San Diego, CA, USA) were employed for statistical analysis and graphic presentations.

A two-way analysis of variance (ANOVA) with repeated measures was performed to localize significant differences. We matched the data by both factors (2 conditions: tDCS vs. sham-tDCS) vs. 10 ReST practice sessions for all comparisons. These results were then analyzed using Bonferroni’s multiple comparisons test.

To estimate whether the brain stimulation procedure could predict speech performance, a linear regression analysis was applied on the percentage (%) of correct instances of speech sound production of trisyllabic NSWs in ReST practice across 10 sessions under tDCS or sham-tDCS conditions. The slopes of linear curves were further compared between conditions. A paired *t*-test was also applied to compare the number of correct utterances of trisyllabic NSWs during ReST practice from the initial and final probe sessions.

P3-CSDs from selected ROIs obtained during ReST practice were analyzed using two-way ANOVAs with repeated measures matched by both factors (2 conditions (tDCS vs. sham-tDCS) vs. 10 sessions), followed by Bonferroni’s multiple comparisons test. Cross-correlations between P3-CSDs from these selected ROIs were also examined.

Linear regressions were applied on the percentage (%) of correct productions of trisyllabic NSWs during ReST practice over the mean of P3-CSDs obtained across 10 sessions under both tDCS and sham-tDCS conditions, as well as between P3-CSDs from the main ROIs.

## 3. Results

### 3.1. Participant

The participant of this study was a 20-year-old male individual with clinical and genetic diagnosis of Trisomy of chromosome 21, fulfilling the criteria for an intellectual disability or intellectual development disorder, and for communication disorders, more specifically for speech sound disorder.

In the absence of a gold-standard test for CAS diagnosis, the clinical identification of CAS was based on the checklist published by Namasivayam et al. [50]. This checklist states that the presence of at least 7 of 12 behavioral features suggests a diagnosis of CAS. The participant of this study presented 10 of the 12 features on this CAS checklist and was remotely diagnosed by one of the co-authors (ACEV), a qualified speech–language pathologist, as having CAS.

Additionally, to stablish the severity of the speech disorder, the Percentage of Correct Consonants (PCC) was calculated [39]. This index is obtained through dividing the Number of Correct Consonants (NCC) by the total number of consonants (NCC added to the Number of Incorrect Consonants (NIC)) and multiplying the quotient by one hundred. Based on the PCC result, the speech disorder is classified into four categories: severe (PCC < 50%), moderate–severe (50% < PCC < 65%), mild–moderate (65% < PCC < 85%), and mild (85% < PCC < 100%) [39]. In the PCC index, omissions, substitutions, and distortions are considered errors [38,39,51]. In this study, the PCC index was calculated from picture naming and word imitation scored in the ABFW test [52]. As per the PCC scores, the participant demonstrated moderate–severe speech disorder (PCC = 61.6%; 52.2% for figure naming and 71% for word naming).

Language comprehension was evaluated using the vocabulary section of the ABFW and MTL-BR at the beginning of the study. In the vocabulary section of the ABFW, the participant correctly verbally identified pictures representing nine different conceptual fields at the typical rate expected for 7-year-old children, which is the maximum age that this test was formulated for. In the MTL-BR test, the participant showed 100% (5 out of 5) word comprehension and 57% phrase comprehension (8 out of 14), giving a total score of 68.4%. Both tests showed that the participant had adequate language comprehension.

Thus, despite intellectual limitation and the severity of the CAS, the participant showed adequate language comprehension and was able to understand Brazilian Portuguese and to carry out the experimental instructions. Moreover, he was in good general health condition, with no other diagnosis of any mental disorders, restrictions for brain stimulation procedures, or past or current illnesses or abnormalities in laboratory tests that could be aggravated during the treatment.

### 3.2. ReST Performance

#### 3.2.1. Speech Sound Production

The percentage of correct responses for sound production during the first, second, and third sessions was larger under the sham-tDCS condition by 8.0, 3.7, and 1.25 times (between 10 and 22%), respectively, over the tDCS condition (between 2 and 8%). However, the performance under tDCS surpassed the sham-tDCS performance after the fourth session, reaching a plateau in the last three sessions, in which the percentages of correct responses (around 40%) were shown to be 2.3, 3.0, and 2.2 times greater than the sham-tDCS condition (around 20%) (Figure 3). The percentage of correct responses for sound production was between 8 and 20% in the four control sessions, while it increased 10-fold in probe test sessions, from 6% in the initial session to 40% at the end of the study protocol (Figure 3A).

The two-way ANOVA with repeated measures was not different in the inter-condition analysis [F(1,49) = 2.72, MSE = 0.71], but it showed a statistically significant difference in the correct utterance of trisyllabic NSWs in ReST practice in the intra-condition analysis [F(9,441) = 5.92, MSE = 0.68, *p* < 0.0001, ω*p*^2^ = 0.0363] and also a significant interaction between factors [F(9,441) = 5.71, *p* < 0.0001, MSE = 0.10, ω*p*^2^ = 0.0315]. Bonferroni’s multiple comparisons test showed statistically significant differences when comparing data from the 4th to 10th sessions to those obtained in the 1st to 3rd sessions under the tDCS condition. No intra-condition differences were found across sessions under the sham-tDCS condition.

Linear regression analysis showed a statistically significant increase in the % of correct responses under tDCS condition [Y = −2.714 + 2.143X, r^2^ = 0.93; F(1,8) = 112.4, *p* < 0.0001]. The slope under the sham-tDCS condition was not statistically significant [Y = 16.38 + 0.0625X, r^2^ = 0.014; F(1,8) = 0.12, *p* = 0.74]. There was a significant difference when comparing slopes under the tDCS and sham-tDCS conditions [F(1,16) = 57.87, *p* < 0.0001] (Figure 3A).

#### 3.2.2. Probe: Pre- and Post-Analysis

A statistically significant increase in the correct utterance of trisyllabic NSWs in ReST performance was observed between the initial and final probes (t = 4.63, df = 49; *p* < 0.0001, paired *t*-test) (Figure 3A).

#### 3.2.3. Consistency of NSWs Utterances

Differences in the consistency of trisyllabic NSW utterances were observed between the tDCS and sham-tDCS conditions. Under the tDCS condition, 4 of 10 NSWs were correctly repeated four or five times in the last two sessions. In comparison, under the sham-tDCS condition, 0 to 1 NSWs were consistently repeated in the last two sessions. No NSWs were consistently repeated in the control sessions, and three NSWs were correctly repeated 4–5 times in the final probe test. A statistically significant upward slope of sound consistency under tDCS condition was clearly shown in the linear regression analysis [Y = −1.15 + 0.213X, r^2^ = 0.83; F(1,8) = 39.7, *p* = 0.0002]. This pattern was not observed in the slope under the sham-tDCS condition [Y = 0.75 − 0.012X, r^2^ = 0.03, *p* = 0.66]. The slope in the sham-tCDS condition was significantly different compared to the slope under the tDCS condition [F(1,16) = 27.98, *p* < 0.0001] (Figure 3B).

### 3.3. Speech Assessments

An analysis of the phonological ABFW showed a significant clinical improvement in figure naming (about 28.9%); the PCC index increased from moderate–severe (PCC = 47 out of 90, i.e., 52.22%) to mild–moderate (PCC = 73 out of 90, i.e., 81.11%). There was no change in severity for word imitation (i.e., it stayed at mild–moderate severity). There was a slight increase (7.5%) in the PCC index from 71.0% (76 out of 107) to 78.5% (84 out of 107).

The FOCUS scores did not significantly change in the study (<9 difference, likely meaning no meaningful clinical change, according to guidelines from the Preschool Speech and Language Outcome Measurement) [53]. The scores of the FOCUS-34 clinician form were 70 and 75 at the initial and final evaluations, respectively, and the scores of the FOCUS-34 parental form were 56 to 60 at the initial and final evaluations, respectively. The FOCUS assesses the participation of the child in a broader social-communication context. We potentially attribute this lack of change in functional communication to the strict social restrictions put in place during the COVID-19 pandemic.

### 3.4. EEG/ERPs

Differences in brain activation were observed during the P3 interval (250–350 ms). The current source densities (CSDs) of the prefrontal region were, in general, reduced under the tDCS condition during the sessions when compared to the sham-tDCS condition (Figure 4, Table 1). This reduction was especially marked from the left side of the brain, including Broca’s area and the ventromedial prefrontal cortex (vmPFC).

An opposite pattern was observed for Wernicke’s area, the left supramarginal gyrus, and Sylvian temporal–parietal junction, regions related to speech motor function (Figure 5, Table 1). These brain regions followed inverted U-shaped curves from sessions 6 to 10 under tDCS condition; meanwhile, there was a reversed pattern with U-shaped curves over these sessions under the sham-tDCS condition (Figure 5).

Interestingly, CSDs from Wernicke’s area under the tDCS condition progressively increased with the improvement of speech sound production over the learning sessions (Figure 6a) [Y = 0.000667 + 0.0002672X, r^2^ = 0.49, F(1,8) = 7.54, *p* = 0.0252, linear regression analysis]. No other brain regions of interest were linearly related to speech utterances during ReST practice.

Additionally, the CSDs from Wernicke’s area progressively increased with increasing CSDs from the supramarginal gyrus from the left hemisphere [Y = 0.003 + 0.407X, r^2^ = 0.56, F(1,8) = 10.28, *p* = 0.0125] and from the Sylvian temporal–parietal junction [Y = 0.00183 + 0.575X, r^2^ = 0.67, F(1,8) = 15.96, *p* = 0.004] under the tDCS condition (Figure 6b). Surprisingly, no other brain region depicted in this study was related to Wernicke’s area activation, not even Broca’s area. There was also no relation found between Wernicke’s area and other brain regions under the sham-tDCS condition (Figure 6c).

## 4. Discussion

The present study investigated the effects of tDCS combined with a motor learning task in developmental apraxia of speech co-existing with T21. Bilateral brain stimulation using tDCS (anodal stimulation of Broca’s area (left IFG) and cathodal stimulation of its homologue contralateral region (right IFG)) progressively increased the accuracy of speech sound production (best performance reached ~40%), indicating a significant clinical gain. In contrast, the performance under the sham-tDCS condition did not change and was around 20% from the beginning to the end of the study protocol. Improvements were also noted for speech consistency and the phonological ABFW (figure naming) test.

Marangolo et al. [11] showed that anodal tDCS over the left IFG (with the cathode over the right supraorbital region) produced long-term speech improvements in three patients with chronic aphasia and apraxia. They observed an increase in the mean percentage of response accuracy from 7.1% to about 34% after five tDCS sessions (1 mA, 35 mm^2^, for 20 min) compared to a change of 18.3% after five sham sessions. Between pre- and post-training, there was a mean difference in the response accuracy percentage of 26.7% for anodal tDCS and 11.7% for the sham condition.

In a follow-up tDCS study on eight patients with chronic aphasia and apraxia, Marangolo et al. [10] demonstrated that active bihemispheric stimulation (2 mA, 35 cm^2^, for 20 min) over the left and right IFG over 10 sessions increased the accuracy of correct words by 22% relative to sham-tDCS.

In the current study, similar gains in speech accuracy were observed. The mean difference between the first and fifth tDCS sessions was 26%, while between the first and fifth sham-tDCS sessions, it was only 6%. The overall accuracy increased by 38% after 10 sessions of tDCS and only by 2% after 10 sessions of the sham treatment. Thus, the gains in speech accuracy produced by tDCS in apraxia of speech in a single individual with T21 resemble those reported by Marangolo et al. [10,11] in adult patients with aphasia and apraxia of speech.

Marangolo et al. [10] observed that tDCS-induced changes generalized to other tasks administered before and after the treatment. A generalization effect was also observed by Themistiocleous et al. [13], as they found that the sounds of untrained words were 47% shorter in the tDCS condition compared to the sham immediately after treatment. In our study, some transference could be inferred through comparing the sound production of an untrained 10-NSW set (probe) applied at the beginning and at the end of the study protocol. The gain of speech sound accuracy was 34% (from the initial 6% to the final 40%). Moreover, some transference could be inferred from the 28.9% increase in the PCC index scores obtained from the ABFW test.

The pattern of Wernicke’s area activity during ReST training seemed to predict the pattern of gain in speech sound accuracy over 10 sessions under the tDCS condition. Changes in Wernicke’s area following speech motor intervention have been reported earlier by Kadis et al. [16]. They investigated cortical thickness changes in response to 8 weeks of PROMPT intervention (a type of speech motor intervention) in children (ages 3–6 years) with CAS. Following therapy, eight of nine children with apraxia demonstrated a significant thinning of the left posterior superior temporal gyrus (canonical Wernicke’s area). They argued that these findings demonstrated experience-dependent structural plasticity in children with CAS. However, in their study, the degree of cortical thinning was not significantly correlated to the change in standardized speech assessments [16].

Much beyond what has been classically conceived as related to language comprehension, the left posterior Superior Temporal Gyrus (pSTG) together with the adjacent supramarginal gyrus, named Wernicke’s area, has recently been considered to be critical for speech production [54,55,56]. According to Binder [52], neuroimaging studies have provided evidence that Wernicke’s area is the general area of the cortex responsible for phonological or speech sound representations, which are essential for speech output.

In the present study, the anodal tDCS, but not sham-tDCS, over Broca’s area (having the cathode placed over the contralateral region) may have triggered the recruitment of Wernicke’s area when the subject was trained to speak trisyllabic NSWs (i.e., with no associated semantic meaning), successively presented in written and audio formats. Furthermore, the activation of Wernicke’s area triggered by Broca’s anodal tDCS was positively correlated to the activation of the supramarginal gyrus in the left hemisphere and of the Sylvian temporal–parietal junction.

Today, the left pSTG and adjacent cortex in the superior temporal sulcus and supramarginal gyrus regions are thought to store and mentally activate phonological (speech sounds) forms, a process termed phonological representation (or phonological encoding, phonological access, phonological retrieval, etc.) (see [55]). This author specifies that “phonological” refers to the spoken form of the word, not the written form or the meaning. According to Binder [55], the phonological representation is a necessary stage prior to all speech output tasks and is also needed to maintain speech sounds in short-term memory. Patients with lesions in the left pSTG and supramarginal gyrus are specifically unable to retrieve an internal mental image of the phonemes represented by written words.

In a series of left hemisphere stroke patients, Pillay et al. [57] identified pre-articulatory phonological representation (phonological access or phonological retrieval) as correlated with damage to a focal region of the cortex and white matter caudal to the posterior Sylvian fissure, including the posterior supramarginal gyrus and adjacent anterior angular gyrus, planum temporale, and pSTG, and no correlation was observed with Broca’s area, the insula, or the sensorimotor cortex. Additionally, they found no correlation between damage in the posterior peri-Sylvian region and spoken word comprehension.

The concept of the phonological representation of speech sound production seems to fit well with the core concepts of the ReST approach. The principles of the (speech) motor learning procedure employed in ReST requires the subject to build a mental image of the spoken forms of NSWs and retrieve them from their verbal short-term memory to produce them correctly. Thus, it may be possible that the ReST task requires the function of Wernicke’s area, triggered by the repetitive anodal tDCS over Broca’s area in the present study. The involvement of Wernicke’s area with the adjacent supramarginal gyrus and superior temporal sulcus, including the Sylvian temporal–parietal junction, may be necessary to the processing of phonological representation of NSWs.

Studies on phonological short-term memory have shown that the posterior end of the Sylvian temporal–parietal area, a region in the posterior portion of the planum temporale, is activated during stimulus encoding (perception) and covert rehearsal [58]. This area seems to be maximally activated during phonological rehearsal tasks, and it has been thought that it would function as an interface site for the integration of sensory and vocal tract-related motor representations of complex sound sequences, including speech and music (see [58]). The Sylvian temporal–parietal region may be critical for the transformation of an auditory input code to an articulatory (or output) code occurring during tests of simple repetition as well as phonological working memory [58].

Ferpozzi et al. [59] suggested that Broca’s area might be involved in cognitive pre-articulatory function (i.e., operating as a functional gate), authorizing the phonetic translation preceding speech articulation executed by the motor areas. In the present study, the activation of Broca’s area was not strong under anodal tDCS during motor speech training. Considering the “functional gate” hypothesis, Broca’s area may have not been the one with the greater change induced by the anodal tDCS. However, its stimulation possibly allowed the recruitment and activation of other regions/mechanisms required by the phono-articulatory apparatus, such as cognitively orchestrating phonological working memory.

Under the tDCS condition, the activity in brain regions that were correlated to speech accuracy, as mentioned above, followed an “inverted U” shape. It mostly reached maximum activation in sessions 7 and 8 and decreased afterwards, in sessions 9 and 10. This pattern of the activation curvature may suggest that these brain regions were increasingly recruited up to the maximum speech accuracy. After reaching a plateau of speech performance, these regions were possibly no longer in demand, leading to a reduction in the resources needed for phonological working memory processing [13]. Ficek et al. [60] observed a lower functional connectivity of stimulated areas (between the frontal and temporal areas in the language network) after repeated anodal tDCS over the left IFG in patients with primary progressive aphasia, which was correlated with an improved performance in language therapy.

Limitations: In this study, we used an N-of-1 randomized study because of the rare co-occurrence of CAS and T21. Although this N-of-1 randomized study was carefully designed and conducted, it is still limited to one single participant. To strengthen the evidence, further replication (potentially in a multi-center clinical trial) with more participants is needed. The generalizability of this study’s findings is also limited because longitudinal data could not be collected due to restrictions put in place during the height of the first wave of the COVID-19 pandemic lockdown in Brazil (in early 2020).

## 5. Conclusions

Multiple sessions of anodal tDCS over Broca’s area (with the cathode over the contralateral region) improved speech sound accuracy, thus increasing the therapy gain, during training with NSWs in the ReST protocol for a young individual with T21 with CAS.

Changes in P3 ERP activity over the Wernicke’s area seems to predict the progressive gain in speech performance seen under the repetitive anodal bihemispheric tDCS condition. Wernicke’s area activation also appeared to predict the activation of the supramarginal gyrus and Sylvian temporal–parietal junction from the left hemisphere. These brain regions are essential for phonological working memory processes to provide accurate speech sound production.

Bearing in mind the need for replication in other young adult individuals with T21, and that a sham-controlled randomized clinical trial with parallel group comparisons with larger sample size needs to be conducted, we may suggest, with a great caution at this moment, that NIBS, such as tDCS, over the speech sound network could be useful to help with the treatment of apraxia of speech in this population.

## Figures and Tables

**Figure 1 brainsci-14-00058-f001:**
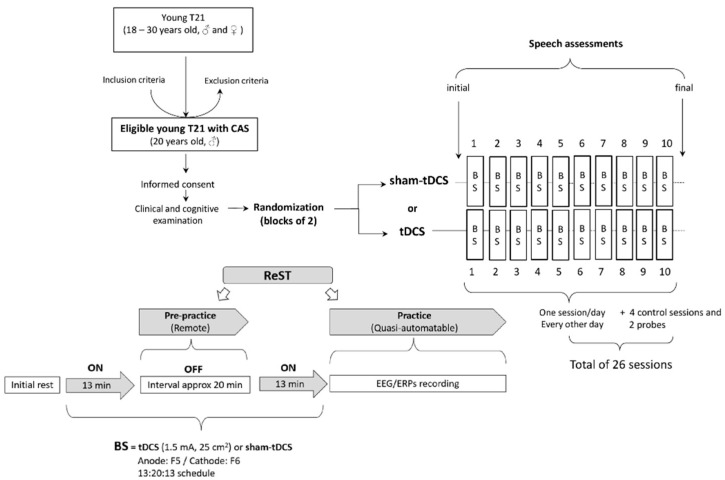
Non-invasive brain stimulation (BS), using tDCS or sham-tDCS, was presented to a participant with CAS and T21 during ReST intervention. A bilateral (left: anode over F5 [Broca’s area—BA44/45] and, right: cathode over F6 [right contralateral region]) tDCS (1.5 mA, 25 cm^2^, in 13:20:13 schedule) stimulation or a sham-tDCS were randomly (blocks of 2) distributed across the intervention block (10 sessions for each condition). Four control sessions were included along with BS sessions. Electrophysiological event-related potentials (EEG/ERPs) and speech production data were acquired during ReST practice phases across all sessions. Speech assessments (probe word data, phonological, vocabulary and global communication) were administered at the beginning (initial) and at the end (final) of the study protocol.

**Figure 2 brainsci-14-00058-f002:**
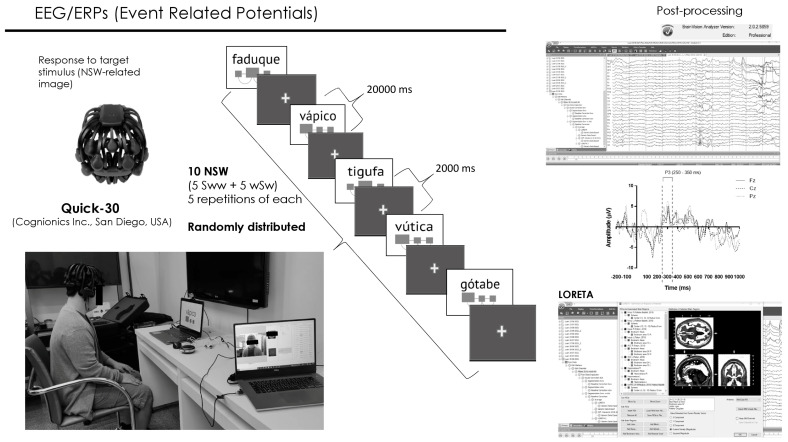
Electrophysiological event-related potentials (EEG/ERPs) registered with Quick-30 EEG system over ReST practice performance having trisyllabic nonsense words (NSWs) with two different accentuation pattern [Strong-weak-weak (Sww) and weak-Strong-weak (wSw)] as cued stimuli. Offline post-processing of collected data with BrainVision Analyzer 2.1.2 Professional software from which current source densities of P3 component segment (250–350 ms) from regions of interest were extracted.

**Figure 3 brainsci-14-00058-f003:**
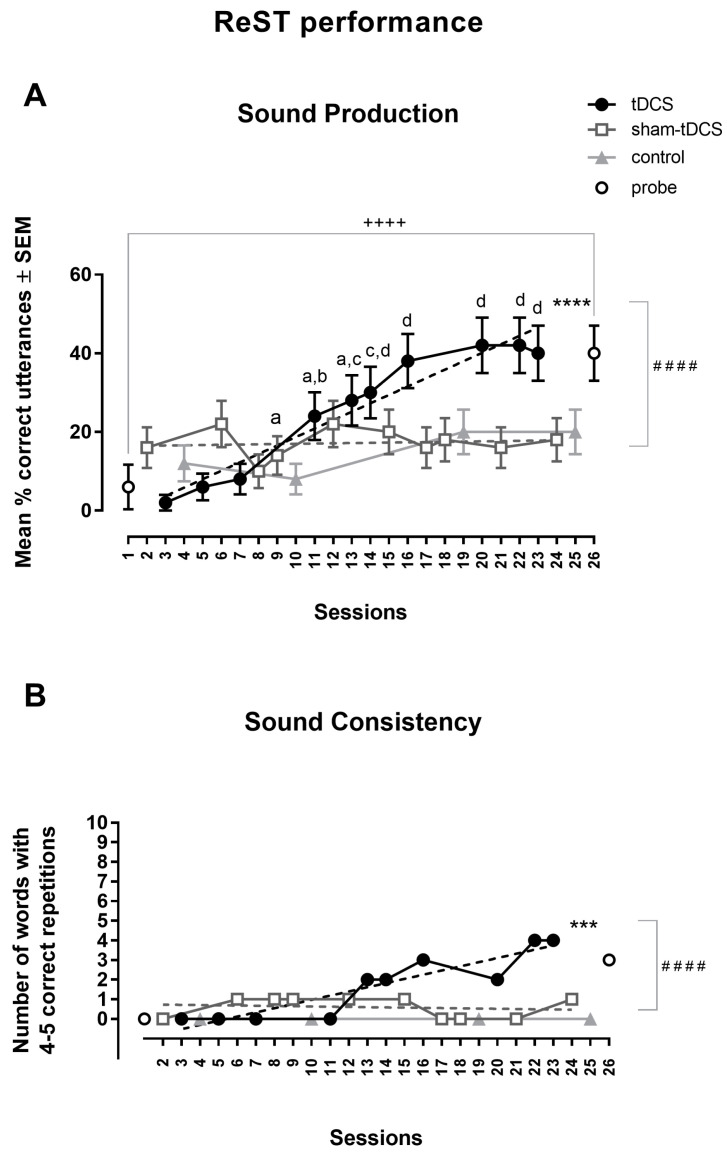
Rest performance. Speech sound production accuracy given by the percentage of correct utterance of trisyllabic nonsense words across sessions (**A**). Sound consistency is given by the number of words with 4–5 correct repetitions in a session (**B**). a = *p* < 0.05, b = *p* < 0.01, c = *p* < 0.001 and d = *p* < 0.0001 (Bonferroni’s multiple comparisons test comparing data from 4th to 10th sessions and those from 1st to 3rd sessions in within-group analysis for tDCS condition only); *** *p* < 0.001; **** *p* < 0.0001 for the slope of the tDCS condition (Linear regression analysis); ^####^ *p* < 0.0001 between slopes of tDCS and sham-tDCS conditions (Linear regression analysis); ^++++^ *p* < 0.0001 (paired *t*-test between probes).

**Figure 4 brainsci-14-00058-f004:**
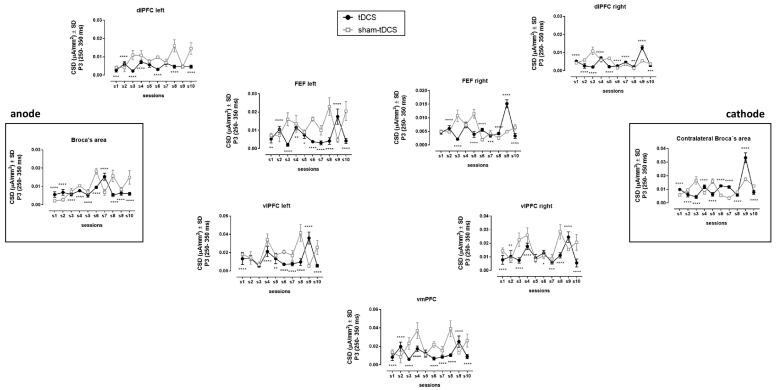
Current Source Densities (CSDs) from left and right prefrontal regions (dlPFC = dorsolateral prefrontal cortex; FEF = frontal eye field; vlPFC = ventrolateral prefrontal cortex; vmPFC = ventromedial prefrontal cortex), surrounding anode (over Broca’s area) and cathode (over contralateral region) electrodes placement, obtained during cognitive potential or P3 (or P300) interval (250–350 ms) through low-resolution electromagnetic tomography (LORETA) analysis of event-related potentials (ERPs) acquired in each ReST practice session conducted under tDCS (1.5 mA, 25 cm^2^, 13:20:13 schedule) or sham-tDCS conditions. * *p* < 0.05, ** *p* < 0.01, *** *p* < 0.001, **** *p* < 0.0001 (Bonferroni’s multiple comparison tests following two-way ANOVAS with repeated measures detailed in Table 1).

**Figure 5 brainsci-14-00058-f005:**
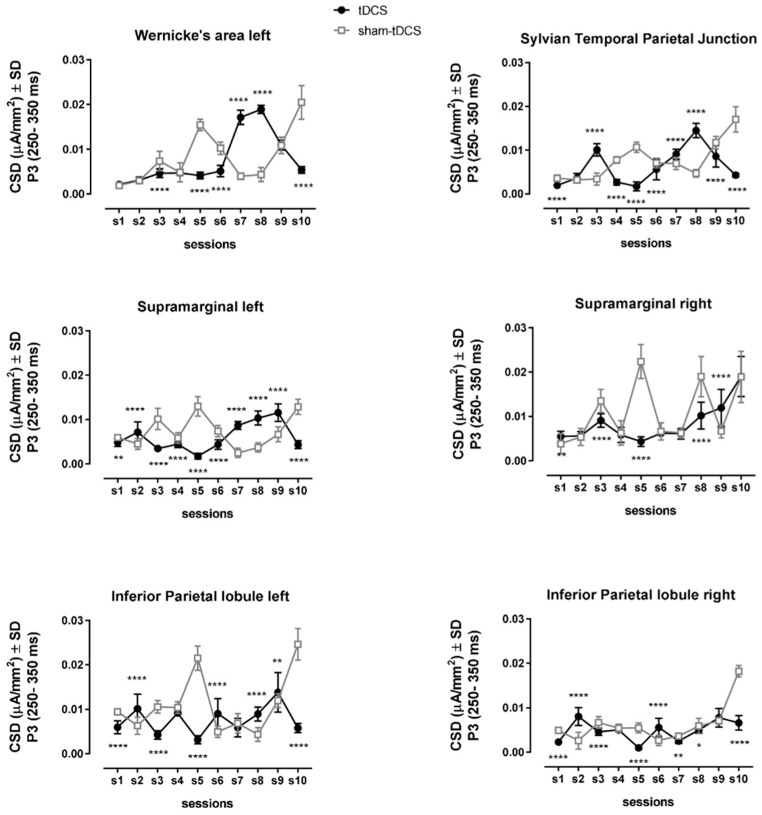
Current Source Densities (CSDs) (µA/mm^2^) from regions related to the speech circuit beyond Broca’s area obtained during cognitive potential or P3 (or P300) interval (250–350 ms) through low-resolution electromagnetic tomography (LORETA) analysis of event-related potentials (ERPs) acquired in each ReST practice session conducted under tDCS (1.5 mA, 25 cm^2^, 13:20:13 schedule) or sham-tDCS conditions. * *p* < 0.05, ** *p* < 0.01, **** *p* < 0.0001 (Bonferroni’s multiple comparison tests following two-way ANOVAS with repeated measures detailed in Table 1).

**Figure 6 brainsci-14-00058-f006:**
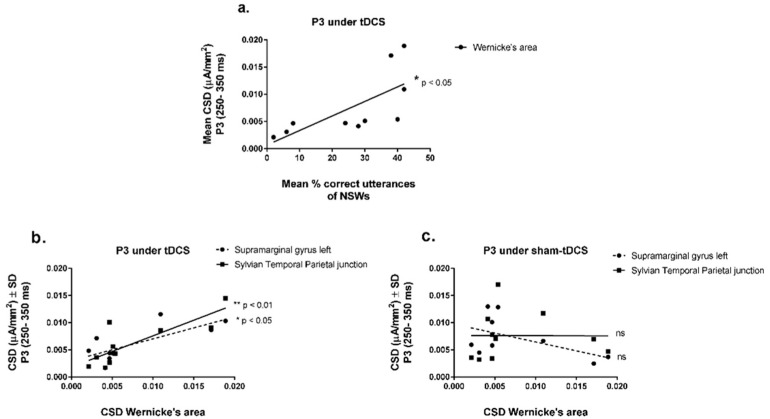
(**a**) Correlation of the mean current source densities (CSDs) (µA/mm^2^) of P3 interval (250–350 ms) from Wernicke’s area (left hemisphere) with the mean % correct utterances of nonsense words (NSWs) of ReST practice under tDCS condition), (**b**) correlations of CSDs of P3 interval from Wernicke’s area (left hemisphere) with regions from speech circuit (supramarginal gyrus and Sylvian Temporal Parietal Junction under tDCS, and (**c**) sham-tDCS conditions. (* *p* < 0.05, ** *p* < 0.01, Linear regression analysis).

**Table 1 brainsci-14-00058-t001:** Comparisons of two-way ANOVAs with repeated measure of P3 interval (250–350 ms) current source densities (CSDs) from brain regions related to the anode and cathode electrodes placement for tDCS or sham-tDCS procedures and to the speech circuit.

Brain Area	Factor	DF	F	MSE	*p* Value	ω*p*^2^	Bonferroni’s Multiple Comparisons Test
Broca (left IFG)	Between condition	(1,49)	244.7	4.1 × 10^−6^	<0.0001	0.048	<0.0001 all comparisons
	Within sessions	(9,441)	359.0	2.8 × 10^−6^	<0.0001	0.430	
	Interaction	(9,441)	415.9	2.3 × 10^−6^	<0.0001	0.407	
Contralateral (right IFG)	Between condition	(1,49)	49.1	2.4 × 10^−6^	<0.0001	0.026	<0.0001 all comparisons
	Within sessions	(9,441)	1642.0	1.7 × 10^−6^	<0.0001	0.576	
	Interaction	(9,441)	716.4	2.6 × 10^−6^	<0.0001	0.374	
dlPFC left	Between condition	(1,49)	5646.0	7.7 × 10^−7^	<0.0001	0.268	<0.001, 0.0001, except sessions 7 and 9
	Within sessions	(9,441)	175.8	2.8 × 10^−6^	<0.0001	0.266	
	Interaction	(9,441)	173.3	3.1 × 10^−6^	<0.0001	0.300	
dlPFC right	Between condition	(1,49)	146.8	4.2 × 10^−7^	<0.0001	0.067	<0.01, 0.001, 0.0001 all comparisons
	Within sessions	(9,441)	789.6	6.1 × 10^−7^	<0.0001	0.470	
	Interaction	(9,441)	555.8	8.4 × 10^−7^	<0.0001	0.450	
FEF left	Between condition	(1,49)	4266.0	1.9 × 10^−6^	<0.0001	0.195	<0.05, 0.01, 0.0001 all comparisons
	Within sessions	(9,441)	79.6	7.9 × 10^−6^	<0.0001	0.133	
	Interaction	(9,441)	324.7	7.4 × 10^−6^	<0.0001	0.505	
FEF right	Between condition	(1,49)	67.3	5.1 × 10^−7^	<0.0001	0.029	<0.001, 0.0001, except sessions 1 and 4
	Within sessions	(9,441)	598.9	7.3 × 10^−7^	<0.0001	0.343	
	Interaction	(9,441)	629.2	1.1 × 10^−6^	<0.0001	0.572	
vlPFC left	Between condition	(1,49)	1071.0	1.1 × 10^−5^	<0.0001	0.087	<0.05, 0.01, 0.0001, except sessions 2 and 3
	Within sessions	(9,441)	163.8	2.5 × 10^−5^	<0.0001	0.283	
	Interaction	(9,441)	274.0	2.4 × 10^−5^	<0.0001	0.456	
vlPFC right	Between condition	(1,49)	1745.0	3.7 × 10^−6^	<0.0001	0.107	*p* < 0.05, 0.01, 0.001, 0.0001, except session 5
	Within sessions	(9,441)	194.5	1.4 × 10^−5^	<0.0001	0.408	
	Interaction	(9,441)	192.5	1.0 × 10^−5^	<0.0001	0.301	
vmPFC	Between condition	(1,49)	1810.0	1.0 × 10^−5^	<0.0001	0.167	<0.0001, except session 5
	Within sessions	(9,441)	154.5	1.0 × 10^−5^	<0.0001	0.257	
	Interaction	(9,441)	179.8	2.5 × 10^−5^	<0.0001	0.382	
Wernicke	Between condition	(1,49)	29.9	3.1 × 10^−6^	<0.0001	0.003	<0.0001, except sessions 1, 2, 4 and 9
	Within sessions	(9,441)	843.2	1.7 × 10^−6^	<0.0001	0.379	
	Interaction	(9,441)	1033.0	1.0 × 10^−6^	<0.0001	0.558	
Sylvian Temporal Parietal junction	Between condition	(1,49)	273.7	1.8 × 10^−6^	<0.0001	0.026	<0.0001, except session 2
	Within sessions	(9,441)	377.1	1.9 × 10^−6^	<0.0001	0.343	
	Interaction	(9,441)	514.8	2.2 × 10^−6^	<0.0001	0.529	
Supramarginal left	Between condition	(1,49)	228.8	1.5 × 10^−6^	<0.0001	0.027	<0.01, 0.0001 all comparisons
	Within sessions	(9,441)	116.6	1.6 × 10^−6^	<0.0001	0.130	
	Interaction	(9,441)	398.9	2.5 × 10^−6^	<0.0001	0.672	
Supramarginal right	Between condition	(1,49)	128.2	1.2 × 10^−5^	<0.0001	0.040	<0.0001, except sessions 1, 2, 4, 6, 7 and 10
	Within sessions	(9,441)	473.6	4.8 × 10^−6^	<0.0001	0.520	
	Interaction	(9,441)	101.3	1.1 × 10^−5^	<0.0001	0.246	
Inferior Parietal Lobule left	Between condition	(1,49)	1164.0	2.6 × 10^−6^	<0.0001	0.094	<0.01, <0.0001, except sessions 4 and 7
	Within sessions	(9,441)	236.7	4.0 × 10^−6^	<0.0001	0.257	
	Interaction	(9,441)	311.3	6.1 × 10^−6^	<0.0001	0.512	
Inferior Parietal Lobule right	Between condition	(1,49)	277.0	1.8 × 10^−6^	<0.0001	0.037	<0.05, 0.01, 0.0001, except sessions 4 and 9
	Within sessions	(9,441)	751.9	1.0 × 10^−6^	<0.0001	0.503	
	Interaction	(9,441)	221.9	2.3 × 10^−6^	<0.0001	0.343	

## Data Availability

The authors confirm that the data supporting the findings of this study are available within the article. However, complementary data/statistical analysis of this study are available on request from the corresponding author, EMN-P. The data are not publicly available due to the diversity and amount of raw data and to assure confidentiality [the diversity and amount of raw data and to assure confidentiality].

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
