# Peer review of "Enhancing Speech Rehabilitation in a Young Adult with Trisomy 21: Integrating Transcranial Direct Current Stimulation (tDCS) with Rapid Syllable Transition Training for Apraxia of Speech"

_brainsci, 2024, doi:10.3390/brainsci14010058_

Round 1

Reviewer 1 Report (Previous Reviewer 4)

Comments and Suggestions for Authors

I really thank the authors for the work they have done and for the topic they have selected. I do believe that in current manuscript type (case report) the work fulfill the criteria in the journal and could of interest for different clinicians.

Author Response

Please see the attachment."

Reviewer 2 Report (Previous Reviewer 2)

Comments and Suggestions for Authors

The authors have done an interesting work. However few questions need to be addressed.

1. Could you elaborate on the selection criteria for the participant with T21 and moderate-severe childhood apraxia of speech (CAS)? Were there any specific characteristics or assessments used for inclusion?

2. What were the limitations observed in this study?

3. In the correlation analysis between speech accuracy under tDCS and Wernicke’s area activation, were there any unexpected or secondary findings observed?

4. Given the positive outcomes observed in this N-of-1 trial, what are the potential implications for future research or clinical applications in treating individuals with T21 and CAS?

5. Considering the small sample size and COVID situation in an N-of-1 trial, how generalizable are these findings to a broader population of individuals with T21 and CAS?

Author Response

Reviewer 3 Report (New Reviewer)

Comments and Suggestions for Authors

Article: [Brain Sciences] Manuscript ID brainsci-2789611 - Review Request  

Enhancing speech rehabilitation in a young adult with Trisomy: integrating transcranial Direct Current Stimulation (tDCS) with Rapid Syllable Transition Training for apraxia of speech

General:

A well written article about the (childhood apraxia of speech). Also as well detailed article, with substantial recent references.  The authors have proposed and adopted Experimental procedures for using use of non-invasive brain stimulation (NIBS) techniques.  The article has been written with rich and detailed about the experiment.  This is very vital.

I found that the authors have relied totally on the statistical results, without indicating any mathematics modeling for this purpose.  I do request to enhance the indicated to statistical analysis, with the mathematical models and descriptions.

No indication of the resulting EEG after the use of non-invasive brain stimulation (NIBS) techniques.  At least to indicate the changing in the measured changes in cortical activation induced by tDCS  through scalp electroencephalography (EEG) using event-related potentials (ERPs). 

What was the ERP in  (2.2.4. Event-Related Potentials in line 233) about ??,  I mean what was the Event-Related Potentials about in terms of experiment.  This needs a description.

Figures; Figure 3, 4, 5, 6,

The language of presentation is fine, … did not detect any issue.

5. Conclusion

It needs to be rewritten to cover aspects of the study and the main outcomes, as both are not clearly stated in the conclusion session.

Author Response

This manuscript is a resubmission of an earlier submission. The following is a list of the peer review reports and author responses from that submission.

Round 1

Reviewer 1 Report

Comments and Suggestions for Authors

The submitted work is case report, and should be changed to article type “case report”. The problem of the present study is the comparison and discussion of the results obtained only on a single subject having trisomy 21  with CAS treated with tDCS, with poststroke patients having CAS and aphasia treated with tDCS, or children with CAS. The genetic nature of this disorder underlying speech problems is evident, but the study gives no control. The study lacks a bigger sample to confirm the findings of the study. Too many speculations are evident in the discussion, looking from the perspective of the study having only a single subject. For example, citing only one other (raw 538) related to Broca’s area, stating that Broca’s area does not have direct control over the phono-articulatory apparatus is a bit questionable. The authors should look deeply into the intraoperative and preoperative mappings of Broca’s area(in neurosurgical settings), and have a clear perspective and evidence that Broca’s area is connected with the M1 region. Also, very hard statements that Wernicke’s do not support language comprehension (raw 493) are questionable (not properly related to the study's aims to state this).

Reviewer 2 Report

Comments and Suggestions for Authors

The authors have performed an interesting and extensive work. However, a few issues need to be addressed.

1.  It is a suggestion. The article title can be reframed as "Enhancing Speech Rehabilitation for Young Adults with Trisomy 21: Integrating Transcranial Direct Current Stimulation (tDCS) with ReST Training for Apraxia Enhancement".

2. The abstract can be more simplified and presented clearly.

3. How many participants were recruited for the study? Do the authors include both male and female? 

4. What criteria were used for participant selection?

5. How did you calculate linear regression? Is there any software employed? 

Comments on the Quality of English Language

The english language can be more simplified.

Reviewer 3 Report

Comments and Suggestions for Authors

This manuscript provides a case study of the influence of transcranial stimulation of the cortex on nonword repetition and the P3 ERP over a 20 session Rapid Syllable Transition Training (ReST) intervention on a 20-year old individual with apraxia of speech and trisomy 21. The study included pre-post behavioral measures of nonword repetition and standardized speech and language assessment and controlled for placebo and practice effects by including sham transcranial stimulation sessions as well as nonword repetition control sessions with no transcranial procedure. The authors found a progressive effect of transcranial stimulation in nonword repetition across sessions that was also reflected in the pre-post behavioral measures. The authors also found a smaller practice effect across the control sessions with no transcranial procedure. The final finding was no improvement across sham transcranial stimulation sessions. 

My only significant comment is for the authors to address the suprising finding of no improvement across sham transcranial stimulation sessions. If the transcranial stimulation procedure is promoting lasting changes through neuroplasticity (as demonstrated in pre-post testing), we would expect some transfer to the sham sessions as well as some level of practice effect for these stimuli (as was found across the control sessions). I would like the authors to explain or at least speculate on the lack of any transfer or practice effect on the set of stimuli used in the sham transcranial stimulation sessions. 

Reviewer 4 Report

Comments and Suggestions for Authors

First of all, I would like to thank the authors for trying to explore the possibility to explore motor speech disorders in Down syndrome population. But, unfortunately I have serious concerns about your work: 

- We do not have information about the subject: mild or moderate ID? neuro imaging features? any comorbidity that could explain apraxia of speech?

- Single case design. Population with DS has a huge heterogeneity in cognitive assessment, even the same person can have a very different performance (without any significant health comorbidities). 

- Speech assessment and other cognitive-functional assessment tools were not validated previously in ADULTS with DS? 

I sincerely do not believe that the work at the present time have enough scientific evidence to be published as an original work. It could be only considered as a case report and for this purpose much more information about clinical status of the patient must be given (personal history: pathological conditions and biographic data, familiar history, neurological physical examination, CAMDEX-DS interview or other similar tools, validated cognitive asssesment, MRI information...). At introduction section you should explain that people with mild and moderate ID do not have aphasic symptoms and that they could properly use two different languages, mainly if they are living in a bilingual territory. 

Round 2

Reviewer 1 Report

Comments and Suggestions for Authors

N/A

Author Response

Many thanks for your time spent reviewing our work. We have made few changes in our conclusions according to the editor's suggestion. We also resubmit it as a case report as you suggested. 

Reviewer 3 Report

Comments and Suggestions for Authors

None

Author Response

Many thanks for your clear understanding of our work. We have made few changes in our conclusions according to the editor's suggestion. We also had to resubmit it as a case report according to the suggestion of other reviewers and the editor. 

Reviewer 4 Report

Comments and Suggestions for Authors

I thank the authors for the modifications they have done. However, I do believe that this work should be considered as a case report and not as a original scientific research work for the same reasons I have explained in the first round of the review, that the authors are not able to solve. 

Author Response

(The authors gave the same response as above.)
